# Contrastive Learning Through Time

**Felix Schneider**
Frankfurt Institute for Advanced Studies
`fschneider@fias.uni-frankfurt.de`

**Xia Xu**
Frankfurt Institute for Advanced Studies, XF-IJRC
`xiaxu@fias.uni-frankfurt.de`

**Markus R. Ernst**
Frankfurt Institute for Advanced Studies
`mernst@fias.uni-frankfurt.de`

**Zhengyang Yu**
Frankfurt Institute for Advanced Studies, XF-IJRC
`zhyu@fias.uni-frankfurt.de`

**Jochen Triesch**
Frankfurt Institute for Advanced Studies
`triesch@fias.uni-frankfurt.de`

## Abstract

Human infants learn to recognize objects largely without supervision. In machine learning, contrastive learning has emerged as a powerful form of unsupervised representation learning. The utility of learned representations for downstream tasks depends strongly on the chosen augmentation operations. Taking inspiration from biology, we here study a framework for unsupervised learning of object representations we call Contrastive Learning Through Time (CLTT). CLTT simulates viewing sequences as they might be experienced by an infant while interacting with objects and avoids arbitrary augmentation operations. Instead, positive pairs are formed by successive views in such unsegmented viewing sequences. Generating viewing sequences procedurally, rather than using natural videos, gives us perfect control over the temporal structure of the input and allows us to ask the following two questions. First, can CLTT approach the performance of fully supervised learning? Second, if so, what are the required conditions on the temporal structure of the input? To answer these questions, we develop a new data set using a near-photorealistic training environment based on ThreeDWorld (TDW). We consider several state-of-the-art contrastive learning methods and demonstrate that CLTT allows linear classification performance that approaches that of the fully supervised setting if subsequent views are sufficiently likely to stem from the same object. We also consider the effect of one object being seen systematically before or after another object. We show that this leads to increased representational similarity between these objects, reminiscent of classic neurobiological findings. The data sets and code for this paper can be downloaded at: https://www.github.com/trieschlab/CLTT.

## 1 Introduction

A hallmark of biological organisms is their ability to learn to understand the world around them in a largely autonomous fashion. Consider learning about visual objects. A human infant is not exposed to object views sampled i.i.d. from some fixed distribution and conveniently labeled image by image, but forms representations of objects and categories during extended interactions with individual objects (Bambach et al., 2018) and requires hardly any (verbal) labels for this (LaTourrette & Waxman, 2019). Mimicking such learning abilities in artificial systems would represent a giant leap forward for artificial intelligence.

3rd Workshop on Shared Visual Representations in Human and Machine Intelligence (SVRHM 2021) of the Neural Information Processing Systems (NeurIPS) conference, Virtual.

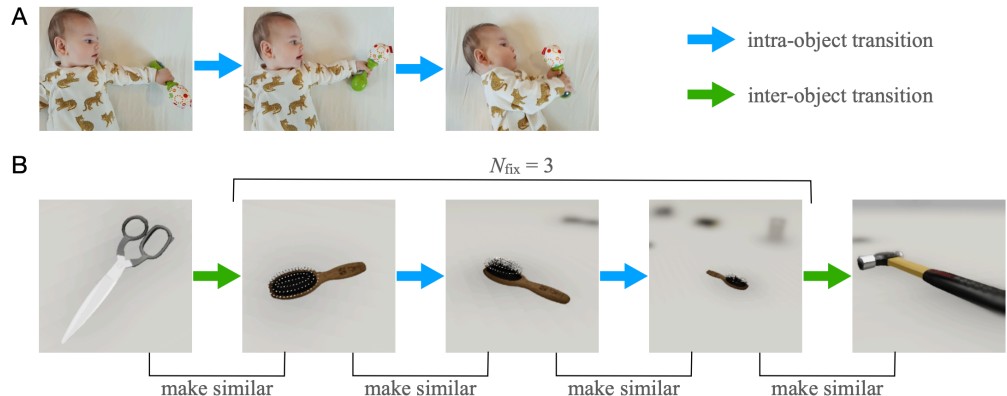

Figure 1: Contrastive Learning Through Time (CLTT). **A.** Infants learn about objects during extended interactions, where they experience different views of an object before directing their attention elsewhere. **B.** Our CLTT approach mimics the essence of such interactions. A certain number $N_{\text{fix}}$ of object views are sampled before directing attention to another object. Latent representations of successive views (both intra-object and inter-object transitions) are made more similar.

Self-supervised learning has emerged as a promising alternative to fully supervised approaches. In the domain of visual object recognition, recent contrastive learning approaches have obtained strong results on standard object recognition benchmarks (Chen et al., 2020a,b; Mitrovic et al., 2021; Grill et al., 2020). These approaches rely on a range of so-called augmentation operations. The basic idea is that an image is transformed through a number of operations (e.g., scaling, flipping, cropping, rotating, blurring, color distortions, pixel noise, ...) that change its appearance but not its meaning (e.g., "cat"). The key mechanism of contrastive learning is to form a representation where such augmented versions of an image are mapped on to close-by latent representations, while at the same time avoiding a "collapse" of the representation, i.e., making sure the network does not simply map all inputs to the same point in the latent space. Not surprisingly, the quality of the learned representations for downstream tasks strongly depends on the chosen augmentation operations (Grill et al., 2020).

Theories of biological learning have also addressed the requirement to learn object representations without (or with only few) labels. The classic theory of how biological organisms learn invariant representations uses the notion of time to substitute for explicit labeling (Földiák, 1991; Rolls & Milward, 2000; Wiskott & Sejnowski, 2002). Biological organisms (including the human infant mentioned above) experience objects across time, typically seeing a sequence of different views of the same object before directing their attention elsewhere (Fig. 1A). Thus, by learning a representation such that subsequent views are mapped onto close-by latent codes (Fig. 1B), a representation should emerge that maps different views of the same object onto similar latent codes, thereby establishing (partial) *invariance*. While this idea has a long history in biological theorizing, it has only recently been explored in a contrastive learning context. The basic idea is to replace the augmentation operations in contrastive learning with natural appearance variation occurring during object interactions. To systematically study this approach, we propose a new Contrastive Learning Through Time (CLTT) framework that permits perfect control over the generated viewing sequences. For our experiments, we utilize the ThreeDWorld (TDW) virtual environment (Gan et al., 2021), which allows near-photorealistic rendering. We also simulate classic biological experiments by Miyashita (1988), demonstrating that objects form similar latent representations in the brain when they are systematically seen one after the other, even if they are visually dissimilar (Miyashita, 1988). We summarize our contributions as follows:

- We develop the CLTT framework using state-of-the-art contrastive learning methods.
- We introduce novel data sets to study CLTT under controlled conditions.
- We systematically analyze the conditions for CLTT to be successful and demonstrate that it approaches fully supervised learning.
- We show that CLTT maps objects that are systematically seen in temporal succession onto similar latent representations, reminiscent of classic neurobiological findings.

## 2   Related Work

An early demonstration that the temporal structure of visual inputs shapes object representations in primate visual cortex was given by Miyashita (1988). He showed 97 images of fractal-like objects to monkeys in always the same order. As the monkeys learned to represent these images in their visual cortices, Miyashita's observations suggested that the representations of objects which were neighbors in the sequence became aligned — even if these objects were visually dissimilar. This effect extended over a few objects. More precisely, his experiments suggested that representations of objects six steps apart in the sequence were still more similar than the average similarity.

Motivated by such findings, there is a long history of neural network and machine learning models exploiting temporal structure for unsupervised representation learning. Földiák (1991) introduced so-called trace learning rules to explain how neurons in the mammalian visual system learn invariance properties, setting a starting point for later models considering multi-layered network architectures (Rolls & Milward, 2000). Another line of research introduced by Wiskott & Sejnowski (2002) explicitly considers the objective of extracting components from an input stream that are slowly changing. A recent variant attempts to do so in a biologically plausible fashion (Lipshutz et al., 2020).

The use of temporal learning objectives in contrastive learning has received increasing attention recently. Among the first, Mobahi et al. (2009) have proposed a method for learning object representations that combines supervised learning and unsupervised learning based on temporal coherence using a siamese neural network architecture. Subsequently, Wang & Gupta (2015) used tracking of patches in videos for unsupervised pre-training. Specifically, they learn an embedding that keeps patches from the same track close in the embedding space.

An approach more closely related to ours has recently been proposed by Orhan et al. (2020). They consider learning on a longitudinal headcam video set from three developing children by Sullivan et al. (2021). They focus on a *temporal classification* approach, where they divide the videos into a finite number of contiguous segments of the same length that they call *temporal classes*. The learning objective is to predict from which of the classes a particular video frame originates. They also consider a temporal contrastive learning objective with the MoCo contrastive learning implementation of Chen et al. (2020b). This objective also aims to make the latent representations of adjacent video frames similar. However, the use of uncontrolled headcam video does not permit titrating the required temporal statistics of the visual input for the approach to work. Another related approach is that of Knights et al. (2020), who learn embeddings of video clips. Their learning objective makes latent codes of adjacent frames within a video clip similar, while making them distinct from latent codes of frames from other video clips. Note that this setup requires the video clips to be segmented, i.e., the system has access to the information where each video starts and ends rather than being exposed to an unlabeled continuous video stream as in Orhan et al. (2020) and CLTT. The same holds true for the recent approach of Feichtenhofer et al. (2021) and also Pan et al. (2021). Finally, the interesting work of Stojanov et al. (2019) has modeled continual infant-like learning of object representations from unsegmented input streams, but they do not consider a contrastive learning approach.

## 3   Methods

**Sampling Sequences of Object Views in CLTT.**   CLTT aims to mimic the stream of object views that an infant may experience during natural interactions with objects, while giving precise control over the statistical properties of this view sequence. Specifically, we here assume that an object is always viewed for $N_{\text{fix}}$ fixations before another object comes into view (we consider stochastic termination of viewing the same object in App. A.6). Importantly, the learner does not have access to the information when a new object comes into view, making the approach fully unsupervised. Generating these view sequences involves two sampling procedures. The first describes how the next view of the *same* object is sampled during the $N_{\text{fix}}$ fixations. We refer to this as *view sampling*. The second describes how the identity of the next object is determined at the end of the $N_{\text{fix}}$ fixations on the same object, which we refer to as *object sampling*.

For the view sampling, we distinguish two sampling methods. Our default method is the *random walk* view sampling. Here, the next view of an object is a "neighbor" of the previous view. For example, in the TDW data set (see below) this corresponds to changing azimuth or elevation by $10°$ or viewing distance by 10 cm. This procedure mimics the infant gradually turning an object or moving around

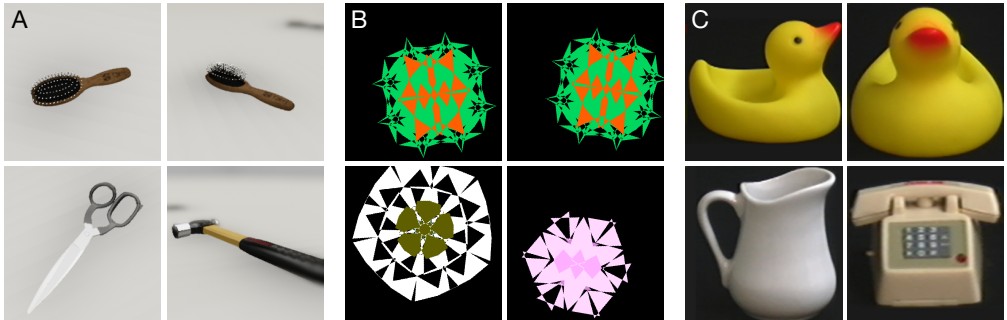

Figure 2: Data sets developed/used in our study. **A.** TDW data set containing common household objects viewed from different orientations and distances. The upper two images show the same example object ("hair brush") from two perspectives. The lower two images show two other example objects **B.** Fractal-like objects inspired by Miyashita (1988). **C.** Example objects from COIL-100.

an object while fixating it. The second method is the *uniform* view sampling. Here, the next view is picked uniformly at random. While this does not mimic infants' viewing sequences, it provides more diversity among successive views, which is expected to aid learning.

For the object sampling we also distinguish two methods. Our default method is the *random order* method. Here, the order of the objects is a new random permutation during each training cycle. This corresponds to the case that the infant encounters objects randomly. The second method uses a *fixed order* of objects, i.e., object A is always followed by object B, etc. in all training cycles. This situation matches the neurobiological experiments by Miyashita (1988) and is expected to lead to an alignment of the latent codes of objects that are consistently seen in succession. Additional details of the sampling procedure are given in App. A.1.

**Contrastive Learning Algorithms for CLTT.** The sampled sequences can be fed into a wide range of contrastive learning methods. Here we consider SimCLR (Chen et al., 2020a) and RELIC (Mitrovic et al., 2021), in which for every positive pair, all the other pairs in a training batch are considered negative pairs. We also experiment with BYOL (Grill et al., 2020), that uses only positive pairs. We refer to the CLTT versions of these algorithms as SimCLR-TT, RELIC-TT, and BYOL-TT, respectively. In all cases, we use a ResNet-18 architecture (He et al., 2016) to transform the input images into 128-dimensional latent representations, which is followed by a two-layer projection head. For the projection head we use batch normalization and a ReLU activation function between the two layers of size 256 and 128. Details of the algorithms and the computational resources are given in App. A.2 and App. A.4.

**Data Sets.** We test these algorithms with three data sets, described in detail in App. A.3. First, we have created a data set of near-photorealistic household objects including depth-of-field effects using the ThreeDWorld (TDW) software (Gan et al., 2021). Second, we have created a dataset of fractal-like images to mimic the biological experiments of Miyashita (1988). Third, to test CLTT with real (rather than computer rendered) images, we use the classic Columbia Object Image Library (COIL-100) data set (Nene et al., 1996).

## 4 Results

**CLTT approaches representation quality resulting from supervised learning.** We first evaluate the proposed family of CLTT methods, namely SimCLR-TT, RELIC-TT, BYOL-TT on the novel ThreeDWorld data set (Fig. 3). We also add a supervised method as baseline, which has access to the true label of each image. We focus on the *random walk* sampling of views and the *random order* procedure for sampling objects. We train the network for 100 epochs and vary $N_{\text{fix}}$ to study its effect. In order to evaluate the quality of learned representations, we use a Linear Least Squares (LLS) classifier to test linear separability.

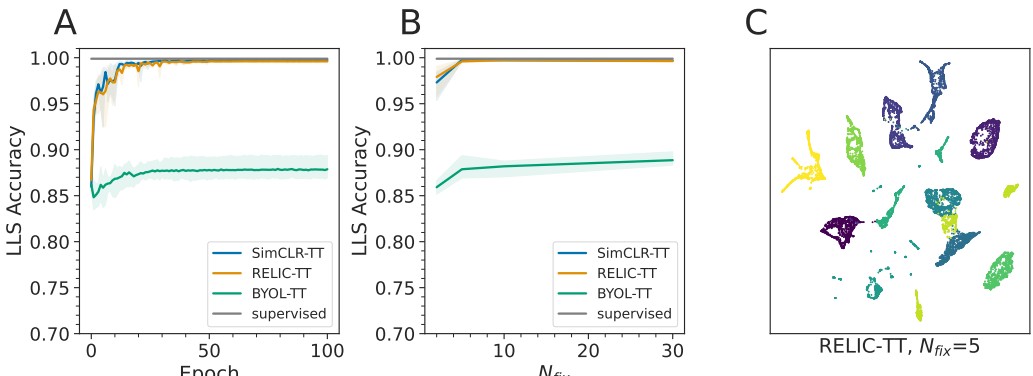

Figure 3: Results for CLTT on the TDW data set. **A.** Comparison of Linear Least Squares (LLS) classification accuracy as a function of training epoch for the different algorithms with $N_{\text{fix}} = 5$. **B.** Final LLS classification accuracy as a function of $N_{\text{fix}}$. **C.** Visualization of the clustering of representations in the latent space using PacMAP, here shown for RELIC-TT with $N_{\text{fix}} = 5$. Each color corresponds to one distinct object. Shaded areas in panels A and B represent the standard deviation based on three individual runs.

In Fig. 3A we show the LLS classification accuracy for $N_{\text{fix}} = 5$ as a function of training epoch. The RELIC-TT method performs best in this setting and approaches the quality of the supervised learning. Figure 3B shows the final LLS classification accuracy after 100 training epochs for different values of $N_{\text{fix}}$. For all methods, greater $N_{\text{fix}}$ tends to improve performance. Figure 3C depicts a PacMAP (Wang et al., 2021) visualization of the latent space resulting from training with RELIC-TT for 100 epochs using $N_{\text{fix}} = 5$. We also perform experiments on the classic COIL-100 data set (Nene et al., 1996) to test the approach with real (non computer-rendered) images (see App. A.5). Experiments with stochastic switching times between different objects are described in App. A.6.

**CLTT aligns latent codes of successively viewed objects.** To relate our framework to biological findings we use the Miyashita-style data set combined with our SimCLR-TT approach. We train the network for $10^6$ stimulus presentations (100 epochs, buffer size = 10,000) and vary $N_{\text{fix}}$. Here, the buffer-size is larger than the number of fractals, because the data set is dynamically generated using fixations, see App. A.3. In line with results from neuroscience, fractals that were presented in succession evoke more similar activations in the latent space than fractals that are far apart in the predefined sequence. Figure 4A depicts the mean cosine similarity between a fractal's latent representation and that of its two $n$-th nearest neighbors along the sequence (in the positive and negative time direction). Note that neighbor zero has a similarity of one, as this corresponds to the identical activation pattern. The curves for all values of $N_{\text{fix}}$ display increased similarity for several nearby neighbors compared to baseline (dashed lines). This indicates that over time not only immediate neighbors, but also more distant fractals become associated (Appendix A.7 provides an intuitive explanation for this effect). In fact, for $N_{\text{fix}} = 2$ we see significant deviations ($p < .01$, two-sided Kolmogorov-Smirnov two-sample test) for the eight next neighbors (in both negative and positive time direction) before merging with baseline. The number of deviating neighbors shrinks with increasing $N_{\text{fix}}$, but significant effects can still be observed. This experiment illustrates a fundamental property of CLTT that can be directly related to biological findings. Figures 4B,C replicate the effect for the TDW and COIL-100 data sets. Note that representations learned with supervised learning do not display any structure of temporal associations and thus similarities do not differ significantly from baseline (not shown).

## 5 Discussion

We have developed a general framework for contrastive learning through time (CLTT). CLTT emulates viewing sequences as may be experienced by infants and uses a temporal contrastive loss that maps subsequent inputs occurring during such interactions onto close-by latent representations ("close in time, will align"). To systematically investigate this approach, we have created a new data set using

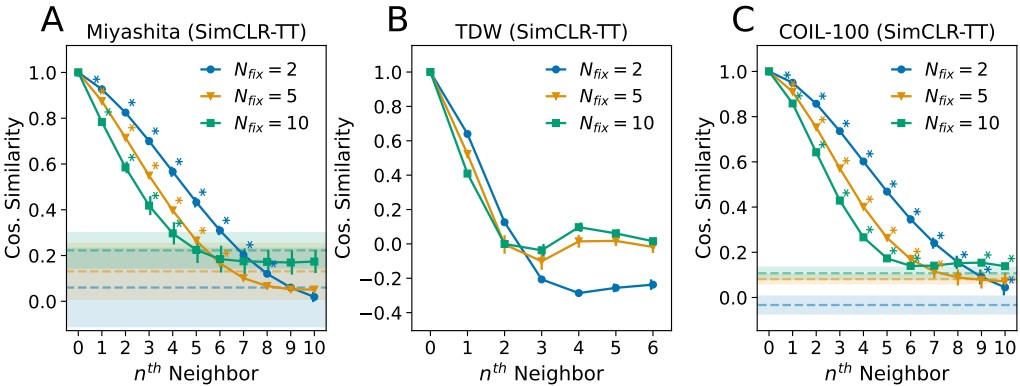

Figure 4: Latent codes of successively viewed objects become aligned. Cosine similarity of latent space representations of objects with different distances along the fixed order object sequence. Different colors show results for different values of $N_{\text{fix}}$. Colored dashed lines and envelopes in panels A and C represent the mean and standard deviation of the neighbors not shown, i.e., $n > 10$. Error-bars depict the standard deviation based on three independent runs. **A.** Miyashita fractals. **B.** TDW objects. **C.** COIL-100 objects.

the ThreeDWorld environment (Gan et al., 2021), allowing us to flexibly simulate different kinds of viewing scenarios in a near photo-realistic fashion. We also validated our approach using a new fractal-like data set inspired by biological experiments and the classic COIL-100 computer vision data set. We have demonstrated that CLTT can approach the quality of representations learned using full supervision. For this, it is important that intra-object transitions (successive fixations fall on the same object) dominate over inter-object transitions (successive fixations fall on different objects). We have also shown that CLTT produces effects reminiscent of classic biological findings suggesting that inputs occurring close in time are mapped onto close-by latent representations by the brain. In App. A.8 we argue that this principle may have much wider applicability.

Our work has a number of limitations. First, while the use of TDW gives us perfect control over all parameters of viewing sequences including scene geometry, sequence of views, lighting conditions, etc., the approach needs to be validated in the real world. Using the COIL-100 data set has been a first step in this direction. Future work should consider first person videos from infants wearing head-mounted cameras (Bambach et al., 2018; Orhan et al., 2020). Second, we have assumed well-separated objects without major occlusions in front of uniform backgrounds. Learning in a cluttered environment is expected to be harder, but it may benefit from foveated vision and additional attentional and figure-ground separation mechanisms, which we plan to incorporate in future work. Third, our eye movement model, which kept gaze on the same object for a certain number of fixations before redirecting it elsewhere, is overly simplistic. In the future, it will be interesting to use more refined models that better reflect (measured) gaze sequences of children and adults. Indeed, it is an interesting question what gaze sequences are particularly beneficial for learning and if and how infants and artificial vision systems can optimize viewing sequences to maximize their own learning progress. This links CLTT to research on intrinsic motivation and (artificial) curiosity. Exploring these issues will bring us closer to building artificial vision systems that can learn truly autonomously.

# 6   Acknowledgements

This research was supported by "The Adaptive Mind" and "The Third Wave of Artificial Intelligence", funded by the Excellence Program of the Hessian Ministry of Higher Education, Science, Research and Art. JT was supported by the Johanna Quandt foundation. XX and ZY thank the XF-IJRC through FIAS for their support. JT thanks Arthur Aubret and Chen Yu for helpful discussions.

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

# A    Appendix

## A.1    Details on the CLTT sampling procedure

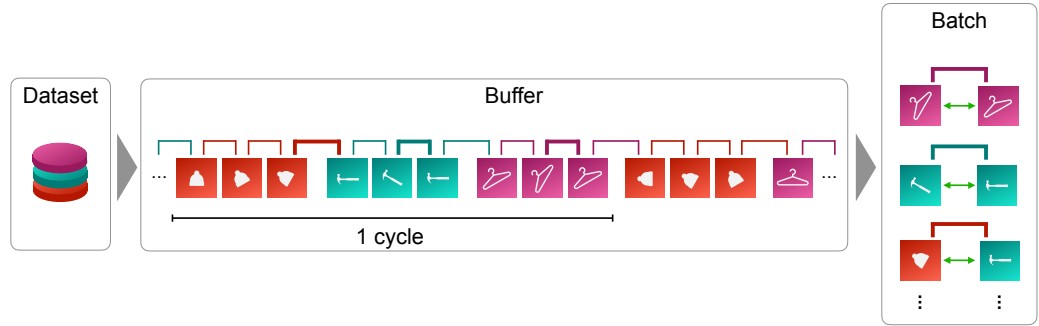

Figure A.1: Sequence generation and sampling procedure. Subsequent object views (same color) are determined by the view sampling procedure, which can either be *random walk* or *uniform*. The number of successive views per object is $N_{\text{fix}}$ (here $N_{\text{fix}} = 3$). Depending on the object sampling method, each cycle either contains the same order of objects (*fixed order*) or a random permutation of objects (*random order*) as shown in the figure. With the resulting sequence of cycles we fill a buffer from which pairs are randomly sampled into a batch (sampled pairs are highlighted with bold brackets). Compare description in main text.

Sampling of object views occurs in cycles. In every cycle, each of the $N_{\text{obj}}$ objects will be chosen exactly once — the order depending on the chosen object sampling method. Thus, a cycle consists of a total of $N_{\text{obj}} \times N_{\text{fix}}$ object views. The $N_{\text{fix}}$ views for each object are determined by the chosen view sampling procedure. Several cycles of $N_{\text{obj}} \times N_{\text{fix}}$ views are stored in a buffer. Here, the size of the buffer equals the size of the underlying data set, but other choices are possible. During learning, batches of pairs of subsequent views are sampled uniformly from the buffer. For our experiments, we use a batch size of 256 image pairs or 512 images in total. The procedure is illustrated in Fig. A.1.

**Statistics of intra-class and inter-class positive and negative pairs in CLTT.**    Successive object views in a viewing sequence form positive pairs during contrastive learning. The sampling procedure introduced in the main text gives rise to two types of positive pairs: *intra-class pairs* and *inter-class pairs*. Intra-class pairs show different views of the same object, whereas inter-class pairs show two different objects. It is useful to establish the relative abundance of different kinds of pairs generated by the CLTT sampling procedure.

The fraction of intra-class pairs is given by $(N_{\text{fix}} - 1)/N_{\text{fix}}$, while the fraction of inter-class pairs is $1/N_{\text{fix}}$). In other words, there are $N_{\text{fix}} - 1$ times as many intra-class pairs than inter-class pairs. In the *random order* method, all possible $N_{\text{obj}}(N_{\text{obj}} - 1)/2$ inter-class pairs occur, whereas in the *fixed order* procedure only $N_{\text{obj}}$ distinct types of inter-class pairs are present.

For a batch with size $N_{\text{batch}}$, there are $N_{\text{batch}}$ pairs containing $2N_{\text{batch}}$ views. For SimCLR-TT and RELIC-TT, we calculate all pairwise similarities between the $2N_{\text{batch}}$ views in the batch, resulting

in a similarity matrix with $4N_{\text{batch}}^2$ entries. Each entry in the matrix denotes either a positive or a negative pair in the loss function, except for the $2N_{\text{batch}}$ diagonal entries. This results in total of $2N_{\text{batch}}$ positive pairs and $4N_{\text{batch}}^2 - 4N_{\text{batch}}$ negative pairs. For positive pairs, the probability of the pair being intra-class is the same as that of sampling an intra-class pair in the batch. Thus, the expected proportion of intra-class and inter-class pairs among the positive pairs are $(N_{\text{fix}} - 1)/N_{\text{fix}}$ and $1/N_{\text{fix}}$, respectively. This results in a total of $2N_{\text{batch}} \times (N_{\text{fix}} - 1)/N_{\text{fix}}$ positive intra-class pairs and $2N_{\text{batch}}/N_{\text{fix}}$ positive inter-class pairs.

For negative pairs, the probability of a certain view forming a pair with another view sampled from the same class is $1/N_{\text{class}}$, where $N_{\text{class}}$ is the number of object classes in the dataset. Thus, the expected proportion of intra-class and inter-class pairs among negative pairs are $1/N_{\text{class}}$ and $(N_{\text{class}} - 1)/N_{\text{class}}$, respectively. Thus, the expected numbers of negative intra-class and inter-class pairs are $\left(4N_{\text{batch}}^2 - 4N_{\text{batch}}\right)/N_{\text{class}}$ and $\left(4N_{\text{batch}}^2 - 4N_{\text{batch}}\right) \times (N_{\text{class}} - 1)/N_{\text{class}}$, respectively.

From the above, we can calculate the ratio between positive and negative intra-class pairs as:

$$\frac{2N_{\text{batch}}(N_{\text{fix}} - 1)/N_{\text{fix}}}{\left(4N_{\text{batch}}^2 - 4N_{\text{batch}}\right)/N_{\text{class}}} = \frac{1}{2}N_{\text{class}}\frac{N_{\text{fix}} - 1}{N_{\text{fix}}}\frac{1}{N_{\text{batch}} - 1}. \tag{1}$$

This ratio grows monotonically with $N_{\text{fix}}$ and the number of classes $N_{\text{class}}$.

For the ratio between positive and negative inter-class pairs we find:

$$\frac{2N_{\text{batch}}/N_{\text{fix}}}{\left(4N_{\text{batch}}^2 - 4N_{\text{batch}}\right)(N_{\text{class}} - 1)/N_{\text{class}}} = \frac{1}{2}\frac{N_{\text{class}}}{N_{\text{class}} - 1}\frac{1}{N_{\text{fix}}}\frac{1}{N_{\text{batch}} - 1}. \tag{2}$$

This ratio decreases monotonically with $N_{\text{fix}}$. Note that both ratios scale with $1/N_{\text{batch}}$. An interesting alternative is to make these ratios independent of the batch size by sampling a fixed number of negative pairs from each batch.

## A.2 Details on the Algorithms

In all our experiments, we use the AdamW optimizer (Loshchilov & Hutter, 2019). The buffer size is determined by the size of the data set. We use a learning rate of $10^{-3}$, which decays by a factor of 0.3 after every 10 epochs.

**SimCLR-TT.** SimCLR (Chen et al., 2020a) is an effective contrastive learning approach for visual representation learning. By defining a wide range of augmentation operations and treating differently augmented images as positive samples, SimCLR has achieved state-of-the-art performance. Its groundbreaking success has triggered a surge of interest in augmentation-based self-supervised learning. For SimCLR-TT we replace the traditional augmentations with successive views as pairs defined above. For each positive pair, all remaining pairs in the batch are considered negative samples. The loss of SimCLR-TT is defined as follows:

$$\mathcal{L}(z_i) = -\log\frac{\exp\left(\text{sim}\left(z_i, z_i'\right)/\tau\right)}{\sum_{k=1, k\neq i}^{N_B}\left[\exp\left(\text{sim}\left(z_i, z_k\right)\right) + \exp\left(\text{sim}\left(z_i, z_k'\right)\right)\right]/\tau}, \tag{3}$$

where $z_i$ and $z_i'$ are the latent codes of a sampled view pair, and $N_B$ is the number of pairs in a batch. Specifically, we use the cosine similarity as the similarity function $\text{sim}(u, v) = \frac{u^\top v}{\|u\|\|v\|}$. For simplicity, we set the temperature parameter $\tau$ from the original SimCLR loss to 1.

**RELIC-TT.** RELIC (Mitrovic et al., 2021) is an approach that uses an additional penalty loss compared to SimCLR. It obtains state-of-the-art results by keeping the similarity distribution of one sample invariant against differently augmented views of other samples. Incorporating this notion into the CLTT approach, we derive RELIC-TT by adding another loss term that can be described as follows:

$$\mathcal{L}_p(z_i) = KL\left(p\left(Y \mid z_i\right), p\left(Y \mid z_i'\right)\right), \tag{4}$$

where $KL$ is the Kullback-Leibler divergence and

$$p\left(Y = j \mid z_i\right) = \frac{\exp\left(\text{sim}\left(z_i, z_j'\right)\right)}{\sum_{k=1}^{N_B} \exp\left(\text{sim}\left(z_i, z_k'\right)\right)}. \tag{5}$$

In the definition above, $z_i$ and $z_i'$ are the latent codes of the first view and the second view of the sampled view pair. The similarities between $z_i$ and all the second views in every pair are calculated and passed to a Softmax function as shown in equation 5. The same computation is done for the second view $z_i'$ for all the other first views, resulting in two probability distributions denoted as $p\left(Y \mid z_i\right)$ and $p\left(Y \mid z_i'\right)$. The penalty loss aims to reduce the KL divergence between these two distributions. The training configuration is the same as used in SimCLR-TT.

**BYOL-TT.** BYOL-TT builds on the Bootstrap Your Own Latent (BYOL) architecture by (Grill et al., 2020). BYOL has been shown to outperform other contrastive learning architectures like SimCLR (Chen et al., 2020a) or MoCov2 (Chen et al., 2020b) on the ImageNet data set (Deng et al., 2009). A key advantage of the BYOL architecture is that it works without negative pairs, which sets it apart from other contrastive learning algorithms. Instead, it uses a second so-called target network. The target network receives an augmented version of the input data like the online network. It will produce a target projection and the online network tries to predict this target projection. The loss function minimizes the similarity between the target projection and the prediction of the online network. In the BYOL-TT implementation, we use a learning rate of $2 \times 10^{-4}$ which decays by a factor of 0.8 after every 10 epochs. The loss is given by:

$$\mathcal{L}_{\theta,\xi} = 2 - 2 \cdot \text{sim}\left(q_\theta(z_\theta), z_\xi'\right) \tag{6}$$

where $q_\theta(z_\theta)$ is the prediction of the online network of $z_\xi'$. $\theta$ corresponds to the weights of the online network and $\xi$ represents the weights of the target network. As similarity function we use again the cosine similarity.

The optimization will be performed with respect to $\theta$, the weights $\xi$ of the target network will be updated using an exponential moving average of the online network where we choose $\tau = 0.996$ as target decay rate. In the original BYOL architecture, an input image $x$ produces two augmented views $v$ and $v'$ which are then shown to the target and online network. In our case no augmentations are applied to the input image, positive pairs are formed by images next to each other in the presented data timeline. Two consecutive images in this timeline will then be used as views $v$ and $v'$.

### A.3 Details on the Data Sets

**ThreeDWorld data set.** This data set was created with the ThreeDWorld (TDW) software (Gan et al., 2021). TDW was chosen, because it enables near-photorealistic rendering including depth-of-field effects and the possibility of simulating physical interaction with objects (which we plan to utilize in the future). Our data set comprises twelve high quality 3D models of common household objects such as a hair brush, a hammer or scissors (compare Fig. 2A). They are arranged in a $4 \times 3$ grid on a white floor. To simulate viewing sequences that an infant may experience while interacting with an object, such as turning an object in hand or moving relative to an object while fixating it, we render object views from different directions and distances. The individual images of the data set can then be arranged into videos simulating different viewing sequences, where an object is seen from different directions and distances. Specifically, to create the different views, we define a spherical coordinate system around the center of each object. Different views are created by changing viewing direction in terms of azimuth (from $0°$ to $350°$ in $10°$ steps), elevation (from $10°$ to $70°$ in $10°$ steps), and distance (0.3 cm to 0.6 cm in 10 cm steps). This gives rise to 1,008 views per object and a total of 12,096 images, which are down-sampled to the size of 64 by 64 pixels. We also created a test set with distinct viewing directions by shifting the azimuth and elevation by $5°$ (from $15°$ to $75°$ and from $5°$ to $355°$ in $10°$ steps). This procedure resulted in a second data set with 12,096 images. Note that due to the varying distances from the objects, sometimes not all parts of an object are visible inside the image and sometimes (parts of) other objects may be present. These appear blurry due to the simulated depth-of-field effect. By sampling different paths through this object, azimuth, elevation and distance space, we generate different viewing sequences that form the input to the CLTT algorithms presented above.

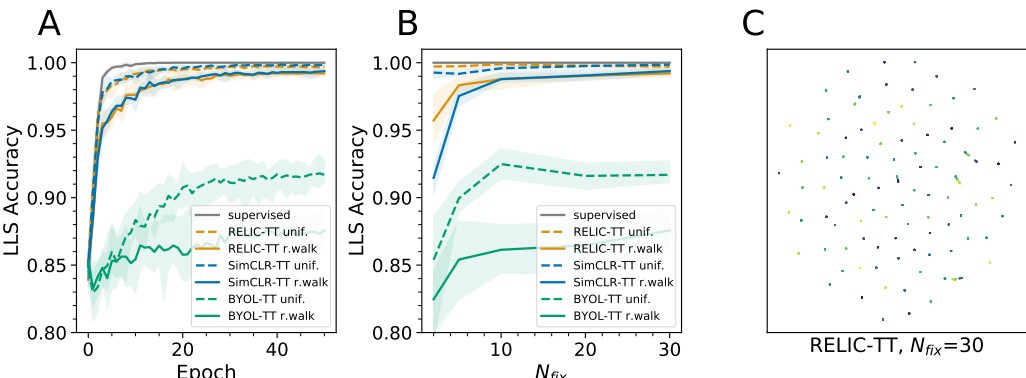

Figure A.2: Results for CLTT on the COIL-100 data set. **A.** Comparison of LLS classification accuracy for the different algorithms and sampling strategies with $N_{\text{fix}} = 30$. **B.** Influence of $N_{\text{fix}}$ on LLS classification accuracy under the two sampling strategies. **C.** Visualization of the clustering of representations in the latent space using PacMAP, here shown for RELIC-TT with $N_{\text{fix}} = 30$ and the *random walk sampling* procedure. Different colors correspond to different objects, but due to the large number of objects, colors have been reused multiple times. Shaded areas in panels A and B represent the standard deviation based on five individual runs.

**Miyashita-style data set.** The Miyashita-style data set draws inspiration from the neuroscience experiments from Miyashita (1988). We adhered to the generation procedure described in (Miyashita et al., 1991) and wrote python code to generate 100 different fractals of size $64 \times 64$. Each of those fractals is unique and highly distinguishable from all others. For our experiments, we give each fractal its own class label and allow for a certain degree of variability to simulate fixation inaccuracies during viewing of the images. Each fixation applies a transformation with a random rotation between $-10°$ and $+10°$, a random rescaling between $90$ and $100\,\%$ and a random translation of up to $15\,\%$ in $x$- and $y$-direction (compare Fig. 2B). The fractals are used exclusively with *fixed order* object sampling, i.e., there is a predefined order of fractals that is repeatedly shown to the network.

**COIL-100 data set.** The Columbia Object Image Library (COIL-100) database (Nene et al., 1996) is composed of color images of 100 objects, each viewed from 72 different directions in $5°$ steps by placing the objects on a motorized turntable making a total of 7,200 images. The objects are seen against a homogeneous black background (compare Fig. 2C). We use 5,400 of these images as the training set and the remaining ones, sampled every $15°$, form the test set.

### A.4 Computational resources

Compared to the original BYOL (Grill et al., 2020), SimCLR (Chen et al., 2020a) and RELIC (Mitrovic et al., 2021) papers, our experiments use smaller data sets and smaller batch sizes. Therefore, we were able to train our models using one GPU of type Nvidia GeForce Titan X, RTX2080Ti or RTX2070Super. Training times for one run ranged from 35 min (Miyashita) to 1:10h (TDW) for a total of 100 epochs.

### A.5 Results with COIL-100 data set

To evaluate CLTT on real (rather than computer rendered) images and a data set with a larger number of objects, we use the classic COIL-100 data set. We compare performance of SimCLR-TT, RELIC-TT, BYOL-TT with a fully supervised method for different values of $N_{\text{fix}}$. We also compare the two sampling strategies: *random walk* sampling where successive views are generated via a random walk in the space of viewing directions and *uniform* sampling where successive views are picked uniformly at random across all possible viewing directions. Figure A.2 shows the LLS accuracy of the different algorithms and sampling strategies as a function of the number of training epochs for $N_{\text{fix}} = 30$. Both SimCLR-TT and RELIC-TT approach supervised performance, although learning is somewhat slower. In contrast, BYOL-TT performs worse. Furthermore, we observe that the *uniform*

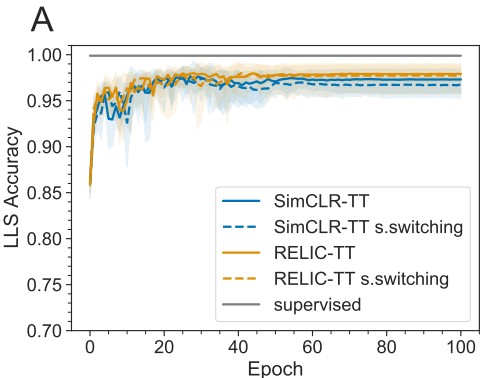 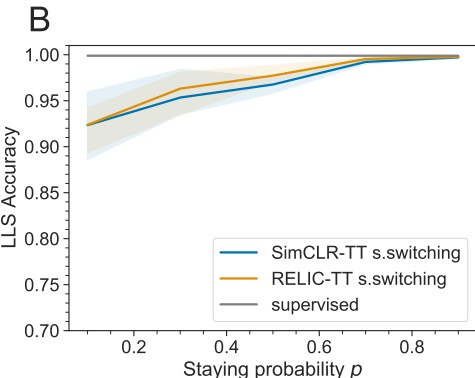

Figure A.3: Results on TDW data set for stochastic switching between objects. **A.** Comparison of LLS accuracy of SimCLR-TT and RELIC-TT with fixed-length for $N_{\text{fix}} = 2$ (solid lines) and stochastic switching for $p = 0.5$, which implies $N_p = 2$ (dashed lines). Random walk sampling is used in all cases. **B.** LLS accuracy of SimCLR-TT and RELIC-TT with random walk sampling for different staying probabilities $p \in \{0.1, 0.3, 0.5, 0.7, 0.9\}$, corresponding to $N_p \in \{1.11, 1.43, 2, 3.33, 10\}$.

sampling leads to better results than the *random walk* sampling. This is not surprising, as it creates more diversity in the training data by allowing for very different views of an object to be grouped as a positive pair during contrastive learning.

Figure A.2B compares the final LLS classification accuracy of the algorithms after 50 training epochs for different values of $N_{\text{fix}}$. For SimCLR-TT and RELIC-TT the performance improves monotonically with $N_{\text{fix}}$ as expected and approaches that of the fully supervised setting. BYOL-TT performs worse and exhibits a large difference between the *random walk* and the *uniform* sampling procedures across all values of $N_{\text{fix}}$.

Figure A.2C visualizes a representative clustering of latent representations with PacMAP for RELIC-TT with $N_{\text{fix}} = 30$ and the *random walk* sampling procedure. The 100 different objects have formed distinct clusters in the latent space.

## A.6 Results for stochastic switching between objects

The sampling procedure described in the main text generates always exactly $N_{\text{fix}}$ successive views of the same object. Here we consider the alternative that the number of successive views of the same object is stochastic, which is more biologically plausible. Specifically, we define a probability $p$ of staying on the same object after every view and a probability $1 - p$ of switching to a different object. This results in variable lengths of sequences viewing the same object. Everything else stays the same. For a given $p$ the expected number of successive fixations on the same object is given by:

$$N_p = \frac{1}{1 - p} \, . \tag{7}$$

This result derives from the expected value of a geometric distribution defined by Bernoulli trials with success rate $1 - p$, which in our case is the switching probability. Conversely, we can calculate $p$ such that $N_p$ equals a certain $N_{\text{fix}}$ as:

$$p = 1 - \frac{1}{N_{\text{fix}}} \, . \tag{8}$$

Figure A.5A compares the stochastic method with the fixed length method for the TDW data set, demonstrating that they perform almost identically. Figure A.5B shows that the staying probability $p$ has a similar effect as $N_{\text{fix}}$: higher values lead to better performance approaching the supervised setting.

## A.7 Additional notes on the alignment of representations of successively viewed objects

Figure 4 shows that the representations of objects that are a few steps apart in the fixed object sequence have heightened similarities compared to baseline, reminiscent of the biological findings of Miyashita (1988). To develop an intuition for why this might be the case, we consider a simplified setting, where the temporal contrastive loss aims to directly align the representations of neighboring objects in the latent space (rather than aligning their projections after the projection head). In SimCLR-TT and RELIC-TT this alignment is achieved by maximizing the cosine similarity of the representations. Maximizing this cosine similarity implies minimizing the *angular distance* between the latent vectors, which is defined as:

$$D_\theta(z_A, z_B) \equiv \frac{1}{\pi} \arccos(\mathrm{sim}(z_A, z_B)), \tag{9}$$

where $z_A$ and $z_B$ are the latent representations of two objects A and B. This is because the angular distance is a strictly monotonically decreasing function of the cosine similarity. Importantly, the angular distance $D_\theta(\cdot, \cdot)$ between two vectors is a metric and therefore obeys the triangle inequality:

$$D_\theta(z_A, z_C) \leq D_\theta(z_A, z_B) + D_\theta(z_B, z_C). \tag{10}$$

Thus, considering the angular distances between the representations of three sequentially presented objects A, B and C, A's representation $z_A$ will be close to its indirect neighbor $z_C$, because $D_\theta(z_A, z_C)$ is upper bounded by $D_\theta(z_A, z_B) + D_\theta(z_B, z_C)$. The upper bounded angular distance implies a lower bounded cosine similarity between the latent representations of A and C. This suggests why not only directly neighboring objects have an increased cosine similarity of their representations but also more distant neighbors. We expect a similar argument to hold for BYOL.

## A.8 Extended discussion of CLTT

The idea that sensory inputs that occur close in time should have similar latent representations may have much broader applicability than the learning of invariant object representations considered here. For example, we conjecture that the "close in time, will align" principle embodied in CLTT could be used, e.g., to learn about color constancy of objects under varying lighting conditions. Furthermore, the related idea of slow feature analysis has been used to model the development of hippocampal-like place cell representations for navigation (Franzius et al., 2007).

Generally, a developing child or a developing AI faces two fundamental problems. It needs to 1) learn about the semantic relationships between things and events in the world and 2) internally represent these relationships. Regarding 2), expressing semantic relationships through alignment of the corresponding internal representations is a simple and elegant solution. Such an organization facilitates generalization and making rapid associations. In particular, it ensures that concepts which are semantically related to a given concept can be found via a *local* search in the latent representation space. Regarding 1), using temporal proximity as an indicator of semantic relatedness appears to be a useful heuristic. Similar to how successive views of an object are compressed into a more abstract and invariant representation of this object by CLTT, it may be useful to compress successive sensorimotor events or even purely internal sequences of "thoughts" into more abstract representations, thereby laying a foundation for (more) abstract thought.

