# OpenReview forum: "Contrastive Learning Through Time"
_NeurIPS.cc/2021/Workshop/SVRHM — SVRHM 2021 Poster_

### Official Review · Reviewer_pYWf · 2021-10-18
**neat idea, interesting preliminary experiments**

**Rating:** 7
**Confidence:** 4

**Review:**

The main idea of this work is that, in human development, agents are exposed to objects via correlated views throughout time, and requiring the representations of these views to be stable might furnish a new type of loss for self-supervised learning approaches. Building on three current methods for self-supervised learning, the authors extend these by treating images nearby in time as positive examples. When the number of contiguous frames grows large enough ($N_\mathrm{fix} = 30$ does best in their experiments), the results can approach those of full supervision.

The idea is interesting, biologically motivated, and I have not seen it proposed elsewhere. There is not much innovation in the technical sense, but the experiments are interesting and worthwhile. The topic is a fit for the workshop, both in content and timeliness. I found the manuscript clear.

A couple of weaknesses worth mentioning:

The method is self-supervised in the sense that it only requires consistency of the embedding, but if I understand correctly, it does require a knowledge of the object labels for proper "windowing" ($W(i)$ in Eq 1). I would be interested in some discussion of mechanisms for producing this in infants.

I realize the space limits are tight, but at least Equation 1 or something similar deserves to be in the main text, just to make the idea more mathematically concrete.

In future work, I would also be curious to see how this method performs on less well-structured data.

Minor points:
- The typesetting of $N_\mathrm{fix}$ varies throughout the manuscript.

---

### Official Review · Reviewer_FbCg · 2021-11-01
**Contributions unclear vs previous work. Findings reported as emergent are built into the methods.**

**Rating:** 4
**Confidence:** 4

**Review:**

Summary:
This work proposes to use temporally close video frames as positive pairs in view-invariant learning in lieu of augmentations of the same image. They evaluate on 3 datasets in various experimental settings and report that "close in time, will align".

Comments:
1. " We develop a general CLTT framework..": It's totally unclear to me what is meant by developing a general CLTT framework. A general framework is never concretely described. Moreover, as acknowledged in the paper, other work has done CLTT before, e.g. Orhan et al. On that note, the related work section could be improved, e.g Feichtenhofer et al, 2021; Puruwalkam and Gupta, 2020; Pan et al 2021. Many previous works have generally tried to make temporally close-by frames have similar representations.
It seems that the main distinction from previous work is that no augmentations are used. However, an important function of augmentations is to prevent trivial shortcuts, this is much less of an issue on a simple data set with only a few distinct objects, with matched backgrounds, no occlusion, no clutter. The proposal to simply use close-by frames and avoid any augmentations will work poorly on more complex video, e.g. see ablations in Feichtenhofer et al, 2021.

2. The claim that the performance can approach that of supervised learning is rather weak. It's unclear what insight one could draw from being on par with supervised learning in this rather simple setting.

3. It is reported that objects presented close-by in time, are found close-by in embedding space as was found in Miyashita, 1988. The presentation of this evaluation as *emergent* is rather confusing to me. Isn't this entirely expected? By *design* positive pairs are optimized to be near each other, whether they are the same object or distinct objects. If A, B form a positive pair and B,C form a positive pair, it can be expected that A,C will be closer to each other than say A, X.

4. In Fig 1B: as N_fix is varied, wouldn't the total amount of training received by the model vary? Aren't the influence of N_fix and amount of training confounded here?

5. Clarity could be improved in the methods sections. e.g. what is a mixing view (l308)? what is it's purpose?

Overall, I'm not sure what message to take away from the paper. To be clear, I'm not saying that it's a bad thing that the data sets are simple. Rather, the flexibility afforded by such simplicity to perform carefully controlled experiments has not really been exploited.

References:
1. Feichtenhofer, Christoph and Fan, Haoqi and Xiong, Bo and Girshick, Ross and He, Kaiming, A Large-Scale Study on Unsupervised Spatiotemporal Representation Learning, CVPR, 2021
2. Senthil Purushwalkam, Abhinav Gupta, Demystifying Contrastive Self-Supervised Learning: Invariances, Augmentations and Dataset Biases, NeruIPS 2020
3. Tian Pan, Yibing Song, Tianyu Yang, Wenhao Jiang, Wei Liu, VideoMoCo: Contrastive Video Representation Learning with Temporally Adversarial Examples, CVPR, 2021

---

### Author Response · Authors · 2021-12-10
**Camera-Ready Paper**

The new version has undergone a major revision. It addresses the reviewers’ concerns and some material has been added.

---

### Decision · Program_Chairs · 2021-11-02

Accept (Poster)